# Budgeted Training for Vision Transformer

**Zhuofan Xia**[1,3*] , **Xuran Pan**[1*], **Xuan Jin**[3*], **Yuan He**[3], **Hui Xue**[3], **Shiji Song**[1], **Gao Huang**[1,2†]

[1]Department of Automation, BNRist, Tsinghua University, Beijing, China
[2]Beijing Academy of Artificial Intelligence, Beijing, China
[3]Alibaba Group, Hangzhou, China

## Abstract

The superior performances of Vision Transformers often come with higher training costs. Compared to their CNN counterpart, Transformer models are hungry for large-scale data and their training schedules are usually prolonged. This sets great restrictions on training Transformers with limited resources, where a proper trade-off between training cost and model performance is longed. In this paper, we address the problem by proposing a framework that enables the training process under *any training budget* from the perspective of model structure, while achieving competitive model performances. Specifically, based on the observation that Transformer exhibits different levels of model redundancies at different training stages, we propose to dynamically control the activation rate of the model structure along the training process and meet the demand on the training budget by adjusting the duration on each level of model complexity. Extensive experiments demonstrate that our framework is applicable to various Vision Transformers, and achieves competitive performances on a wide range of training budgets.

## 1 Introduction

Benefited from the large model capacity, Vision Transformers (ViTs) (Dosovitskiy et al., 2021) have demonstrated their predominant performance on various vision tasks, including object detection (Wang et al., 2021a; Liu et al., 2021; Li et al., 2022b), semantic segmentation (Zheng et al., 2021; Strudel et al., 2021), video understanding (Fan et al., 2021; Arnab et al., 2021), etc. However, these improvements come at huge training costs in which the datasets, the model parameters, and the computation complexities have grown enormous in size. For example, ViT-G/14 with Greedy Soup (Wortsman et al., 2022) achieves 90.9% accuracy on the ImageNet (Deng et al., 2009) benchmark while having 1843M training parameters and being pretrained on a dataset of 3 billion scale. Under this circumstance, computation resource has been becoming an inevitable overhead that prevents common users from training desired vision models.

The methodology of designing modern Transformers is finding the best trade-off between the computation costs and the model performances (Han et al., 2022). Besides the widely used factors like the number of the learnable parameters, the floating point operations (FLOPs) and the inference latency, training cost is also an essential resource that involves training schedule (Wu et al., 2020; Yin et al., 2022; Wang et al., 2022b), memory usage (Pan et al., 2021; Wang et al., 2021c; Ni et al., 2022) and training-stage complexity (Zhang & He, 2020; Gong et al., 2019; Dong et al., 2020). Therefore, the topic of training Transformers efficiently has received broad research interests, especially considering the large-scale data and prolonged schedules in training.

Considering that many of the research labs and companies are not able to afford the full training schedule of the *best* model, one usual solution is to train a *better* one given a desirable and acceptable total training cost. Previous works that focus on addressing the training efficiency problem mainly learn model-specific schedules based on handcraft designs (Gong et al., 2019; Gu et al., 2020; McDanel & Huynh, 2022) or Automated Machine Learning (Li et al., 2022a). However, these approaches either adjust the costs in the training process, or only provide training schedules based on a sparse set of training costs. The inflexibility hinders from generalizing to a pre-defined budget.

---

[*]Equal contribution.
[†]Corresponding author.

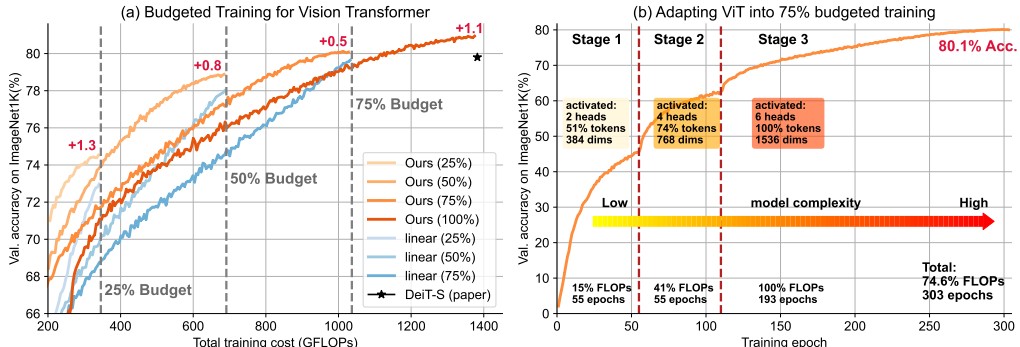

Figure 1: (a) demonstrates our method consistently outperforms Linear-LR (Li et al., 2020) on DeiT-S (Touvron et al., 2021) under three different training budgets of 25%,50%, and 75%. Our method even improves 1.1% over the original model under full budget. (b) shows that our method dynamically adjusts the activation rate of model computation by gradually increasing the attention heads, the token numbers and the MLP hidden dimensions. Our method manages to control the model redundancy during training to meet the given budget while achieving good performance.

In this paper, we take a step forward and focus on the problem of budgeted training (Li et al., 2020), *i.e.,* achieving the highest model performance under **any** given training budget that can be measured by total training time or computation cost. Different from previous work including using smaller model variants, coreset selection (Mirzasoleiman et al., 2020; Killamsetty et al., 2021), efficient training schedules (Li et al., 2020; Chen et al., 2022a), we target this problem from the perspective of the inherent properties of Vision Transformers. Specifically, we focus on leveraging the redundancies of model structure during ViT training. There exists several types of redundancies including the feature diversities across different attention heads, the hidden dimensions in the MLP blocks and the number of attended visual tokens. It is shown that these redundancies are correlated with training process, especially they tend to be higher at early stages.

This motivates us to dynamically control the activation rate of the model along the training process, where less parameters participate in the early training stages and the full model capacity is activated at late stages. As depicted in Fig. 1(b), we activate 2 attention heads, 51% tokens and 384 MLP hidden dimensions in the first stage, which condenses the model redundancy and keeps a low computation cost, and then the activation rate of the model then gradually increases as training goes on. In this way, the training process becomes more compact, where information loss is greatly avoided and results in limited influence on the model performance. Based on this technique, we can adjust the duration at different level of training stages in order to accommodate to different training budgets. Fig. 1(a) shows our method consistently outperforms the baseline at three different budgets. Extensive experiments demonstrate that our method significantly outperforms the other budgeted training baselines and achieves competitive training cost-performance trade-off on various Vision Transformer models.

## 2 RELATED WORKS

**Vision Transformer** (Dosovitskiy et al., 2021) firstly introduces the Transformer (Vaswani et al., 2017) model into the vision tasks. Wang et al. (2021a; 2022a); Liu et al. (2021; 2022); Zhang et al. (2021a) and Li et al. (2021) incorporate the pyramid model architecture with various efficient attention mechanisms for vision. Following the isotropic design, Yuan et al. (2021) leverages overlapped attention, Touvron et al. (2021) benefits from the knowledge distillation (Hinton et al., 2015) and Jiang et al. (2021) proposes the token-labeling technique to improve the data efficiency of the Vision Transformer. Wu et al. (2021); Xu et al. (2021); Dai et al. (2021); Guo et al. (2022) and Pan et al. (2022) make further efforts on combining attention and convolution for the stronger inductive biases.

**Redundancies in Transformers** have been widely studied in the area of NLP. Michel et al. (2019); Zhang et al. (2021b); Voita et al. (2019) prune the redundant attention heads to improve the model efficiency. Bhojanapalli et al. (2021) takes a step further to reuse attention maps in subsequent layers to reduce computations. In vision tasks, the redundancy among visual tokens are of high interest. Wang et al. (2021b) finds that easy samples can be encoded with less tokens, while Xia et al. (2022) adaptively focuses on the most informative attention keys using deformable attention mechanism.

Rao et al. (2021); Xu et al. (2022); Yin et al. (2022); Liang et al. (2022); Song et al. (2021) explore the redundancies in visual tokens by either pruning tokens or introducing sparse computation.

**Budgeted Training** (Li et al., 2020) focuses on training models under certain computational resources constraints by linear decaying learning rates. REX (Chen et al., 2022a) proposes an improved schedule in a profile-sampling fashion. Importance sampling methods (Arazo et al., 2021; Andreis et al., 2021; Killamsetty et al., 2021; Mirzasoleiman et al., 2020) choose the most valuable subset of training data to reduce the training costs with curriculum learning. Pardo et al. (2021) aims to sample informative subset of training data in weakly-supervised object detection. As for the AutoML, Gao et al. (2021) design an automated system to schedule the training tasks according to the budgets. Li et al. (2022a) searches for the optimal training schedule for each budget. Chen et al. (2022b); McDanel & Huynh (2022) can adjust the number of visual tokens during training to adapt to different budgets.

## 3 REDUNDANCIES DURING TRAINING ViTs

### 3.1 BACKGROUND OF VISION TRANSFORMER

We first revisit the architecture of the Vision Transformer (ViT) (Dosovitskiy et al., 2021). As the variant of the Transformer (Vaswani et al., 2017), ViT divides the input $H \times W$ image into a sequence of $p \times p$ patches in the length of $N = HW/p^2$, and embeds them with linear projections into the $C$-dimensional subspace. After prepended with a class token and added by a set of position embeddings, the patch embeddings go through subsequent ViT blocks to extract features. The basic building block of ViT consists of a multi-head self-attention (MHSA) layer and a multi-layer perceptron (MLP) layer. The visual tokens are sequentially processed by every block, with LayerNorm (Ba et al., 2016) and residual connections. Let $\mathbf{z}_l \in \mathbb{R}^{N \times C}$ be the output tokens from the $l$-th block, the block of ViT is formulated as

$$\mathbf{z}'_l = \mathbf{z}_l + \mathrm{MHSA}(\mathrm{LN}(\mathbf{z}_l)), \quad \mathbf{z}_{l+1} = \mathbf{z}'_l + \mathrm{MLP}(\mathrm{LN}(\mathbf{z}'_l)). \tag{1}$$

The MHSA is introduced to learn different representations from separate attention heads. Let $\mathbf{q}, \mathbf{k}, \mathbf{v} \in \mathbb{R}^{N \times C}$ be the query, key and value tokens projected by learnable weights $\mathbf{W_q}, \mathbf{W_k}, \mathbf{W_v} \in \mathbb{R}^{C \times C}$, the attention of the $m$-th head among $M$ heads is computed as

$$\mathbf{h}^{(m)} = \mathrm{SOFTMAX}\left(\mathbf{q}^{(m)}\mathbf{k}^{(m)\top}/\sqrt{d}\right)\mathbf{v}^{(m)}, \tag{2}$$

where $d = C/m$ is the dimension of each head. And then the attention heads are concatenated together to produce the final output, followed by a linear projection $\mathbf{W_O} \in \mathbb{R}^{C \times C}$, written as

$$\mathbf{z} = \mathrm{CONCAT}(\mathbf{h}^{(1)}, \mathbf{h}^{(2)}, \ldots, \mathbf{h}^{(M)})\mathbf{W_O}. \tag{3}$$

The MLP is implemented as two fully-connected layers denoted as $\phi_1(\cdot)$ and $\phi_2(\cdot)$, with a GELU (Hendrycks & Gimpel, 2016) non-linearity inserted between them. The $\phi_1(\cdot)$ takes in the tokens with $C_1$ dimension and projects them to a subspace with dimension $C_2 \geq C_1$. After the activation, these tokens are projected back to $C_1$ dimension as the output of the MLP. We denote the proportion $\gamma = C_2/C_1$ as the **MLP ratio** in the remaining paper. In this section we opt a representative ViT architecture, DeiT-S (Touvron et al., 2021), to analyze the redundancies in ViT training. The same conclusions can be generalized to other Transformer architectures.

### 3.2 REDUNDANCIES IN ATTENTION HEADS

There are six heads in each MHSA layer of DeiT-S, where each head holds a 64-dim representation to compute self-attention. A question will be raised: are all the heads important? In the natural language processing (NLP), it has been observed that zeroing out the representations of some heads or even keeping only one head active is enough to preserve the model performance at the test time (Michel et al., 2019). Bhojanapalli et al. (2021) finds that the attention maps of heads share high similarities between consecutive layers in both vision and NLP tasks. However, few studies inspect this question from the perspective of training.

We compare the CKA similarity between every pair of heads to assess their redundancies. CKA is short for centered kernel alignment (CKA) (Kornblith et al., 2019; Raghu et al., 2021; Cortes et al.,

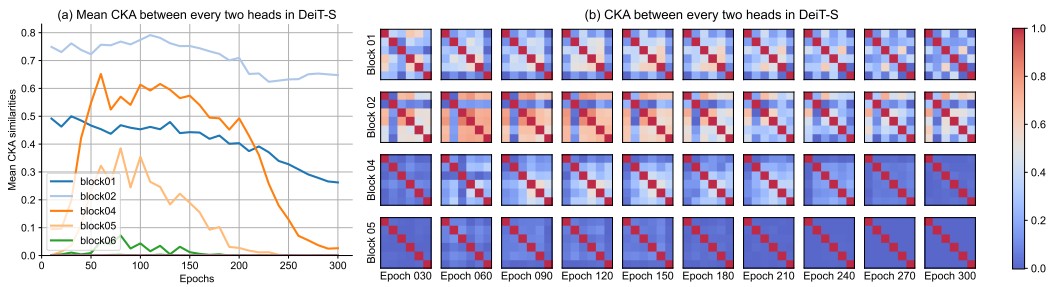

Figure 2: (a) plots the average CKA similarities of all pairs of different heads in each block, where CKA in the shallow blocks (01 ~ 06) decreases as training goes on while the other deep blocks keep a very small value close to zero throughout all epochs. (b) depicts the CKA similarities of the six heads in the shallow blocks in detail. Each square represents a similarity matrix between the features $\mathbf{h}$ of every two heads, where a color close to red indicates a high magnitude and blue indicates a low one.

2012), which is a popular metric to quantize the similarity between the representation of each head. CKA takes two activation matrices $\mathbf{X} \in \mathbb{R}^{m \times p_1}, \mathbf{Y} \in \mathbb{R}^{m \times p_2}$ in the networks as input, and computes the Gram matrices $\mathbf{L} = \mathbf{X}\mathbf{X}^\top, \mathbf{K} = \mathbf{Y}\mathbf{Y}^\top$. By centering the Gram matrices by $\mathbf{H} = \mathbf{I} - \frac{1}{m}\mathbf{1}\mathbf{1}^\top$, $\bar{\mathbf{K}} = \mathbf{H}\mathbf{K}\mathbf{H}, \bar{\mathbf{L}} = \mathbf{H}\mathbf{L}\mathbf{H}$, the Hilbert-Schmidt independence criterion (HSIC) (Gretton et al., 2007) and the CKA are computed as

$$\mathrm{CKA}(\mathbf{K}, \mathbf{L}) = \frac{\mathrm{HSIC}(\mathbf{K}, \mathbf{L})}{\sqrt{\mathrm{HSIC}(\mathbf{K}, \mathbf{K})\mathrm{HSIC}(\mathbf{L}, \mathbf{L})}}, \quad \mathrm{HSIC}(\mathbf{K}, \mathbf{L}) = \frac{\mathrm{vec}(\bar{\mathbf{K}}) \cdot \mathrm{vec}(\bar{\mathbf{L}})}{(m-1)^2}. \tag{4}$$

We compute the $\mathrm{CKA}(\mathbf{h}_l^{(i)}, \mathbf{h}_l^{(j)})$ for each pair of the $i$-th and the $j$-th head in the $l$-th block of the DeiT-S model on ImageNet validation set, where $\mathbf{h}_l^{(i)}$ is the output feature of every separate head attention in Eq.(2) before the concatenation and projection in Eq.(3).

As shown in Fig. 2(a), there are significant descents of these similarities in some multi-head attention blocks. For example, the CKA score in block04 first grows over 0.6 and rapidly decreases to nearly zero as training goes on. To show this trend more clearly, Fig. 2(b) displays the dynamics of similarity matrices at different epochs. From these observations, there exist considerable redundancies between the attention heads and the redundancies are decreasing during the training, which enlightens us to activate fewer heads during the early stage of the training.

## 3.3 REDUNDANCIES IN MLP HIDDEN DIMENSIONS

The dimension expansion and contraction design in the MLP layer of ViTs extracts features by projecting and activating them in a higher-dimensional space, which brings in a great deal of redundancies in features. The searched optimal architectures in ViT-Slim (Chavan et al., 2022) also reveal this type of redundancy in which lots of the hidden dimensions in MLP layers are removed. In terms of training, we leverage the principal component analysis (PCA) to analyze these redundancies among the features in the expanded dimension space projected by $\phi_1(\cdot)$ in the MLP block.

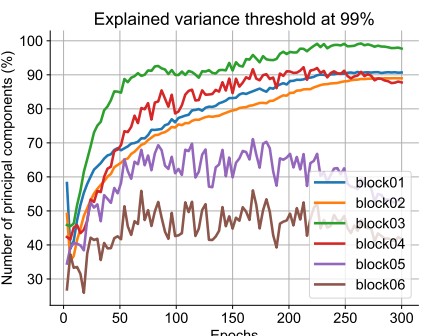

For each token $\phi_1(\mathbf{x}) \in \mathbb{R}^{N \times C_2}$ in the MLP block of DeiT-S, we compute its number of principal components that hold a given proportion of the explained variance. This criterion has been also adopted in pruning redundant neurons in deep networks (Casper et al., 2021). We choose the 99% threshold to measure the alternation of the principal components distribution during training. We plot the numbers of principal components w.r.t. the training epochs in Fig. 3. It is observed that the numbers of princi-

Figure 3: We show the number of principal components of $\phi_1(\mathbf{x})$ which hold the explained variances at a 99% threshold. The numbers of principal components of each block are normalized to a percentile. These numbers of components are averaged across all the examples in the ImageNet validation set.

pal components are growing as the training epoch increases in the shallow and middle blocks. This

phenomenon indicates that in the early training stages, only a few components can hold the most explained variances, *i.e.,* support the projected space. Therefore the redundancies in the early stages of training are in high degree, especially in the shallow blocks. For the deep blocks there also exist the trends of increase of the numbers of components, however they are relatively noisy so we omit them in the figure.

This observation demonstrates that the features are highly linearly dependent in early epochs, lacking diversities among dimensions. As the training goes on, the redundancies are gradually declining, achieving a final state with much fewer redundancies. It suggests that a large MLP ratio $\gamma$ with high hidden dimensions at the early training stage be excessive and a growing $\gamma$ from a smaller value would be beneficial to alleviating this redundancy.

### 3.4 REDUNDANCIES IN VISUAL TOKENS

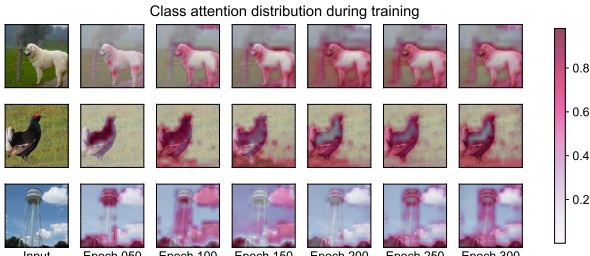

The spatial redundancies in the visual tokens have been widely studied. Many efficient Vision Transformers have sought to eliminate the redundancies during inference time (Wang et al., 2021b; Rao et al., 2021; Yin et al., 2022; Xu et al., 2022; Liang et al., 2022). We follow this line of research to investigate the redundancies among tokens during the training procedure, leveraging the visualization of the class attention. The class attention score is defined as the attention between the class token $\mathbf{q}_{\text{cls}}$ and other patch tokens $\mathbf{k}$, *i.e.,* $\mathbf{A}_{\text{cls}} = \text{softmax}(\mathbf{q}_{\text{cls}}\mathbf{k}^{\top}/\sqrt{d})$. If we view these

Figure 4: Each row displays the evolution of the class attention distribution of the input image. The class attentions are averaged by six heads in a shallow block of the DeiT-S, while the distributions in deeper blocks are nearly uniform. Deeper color indicates a higher attention score.

attention scores as a distribution, the patch tokens with high scores contribute more to the class token, which indicates the informative tokens of the important image patches. As illustrated in Fig. 4, we observe that the patch tokens with higher attention scores first emerge in a small area of the image. And then more patches on the target object begin to get a higher class attention score during training, indicating there are some redundancies in the patch tokens at the early stage of training. This phenomenon exists in some shallow blocks while the class attention in the deep blocks displays a nearly uniform distribution among all visual tokens. Since the effective receptive field of ViTs grow very quickly as the block goes deeper (d'Ascoli et al., 2021; Raghu et al., 2021), the class attention degenerates to a global-pooling operation to aggregate all the tokens. This trend appeals to us to reduce the number of visual tokens in early epochs.

## 4 BUDGETED TRAINING FOR VISION TRANSFORMER

Different from most of the previous works that improves training efficiency based on a fixed schedule, our target is to propose a flexible training framework that can easily adapt to a wide range of training budgets while maintaining competitive performances. In the following, we would first define the problem of budgeted training and then show how we solve the problem by leveraging the redundancies of Vision Transformers.

### 4.1 BUDGETED TRAINING

The total cost of training a deep neural network $\mathcal{M}$ can be defined as the accumulation of the model's computation cost over all the training steps. The budgeted training problem is to minimize the objective loss such that the total computation costs $\mathcal{C}(\mathcal{M})$ of training are under the given budget constraint $\mathcal{B}$,

$$\min_{\boldsymbol{\Theta}} \mathcal{L}(\mathcal{M}; \mathcal{D}), \quad \text{s.t. } \mathcal{C}(\mathcal{M}) = \int_{0}^{T_{\max}} \mathcal{C}(T; \mathcal{M}) \, \mathrm{d}T \leq \mathcal{B}, \tag{5}$$

where $\mathcal{C}(T; \mathcal{M})$ is the computation cost of the model $\mathcal{M}$ at the $T$-th training epoch.

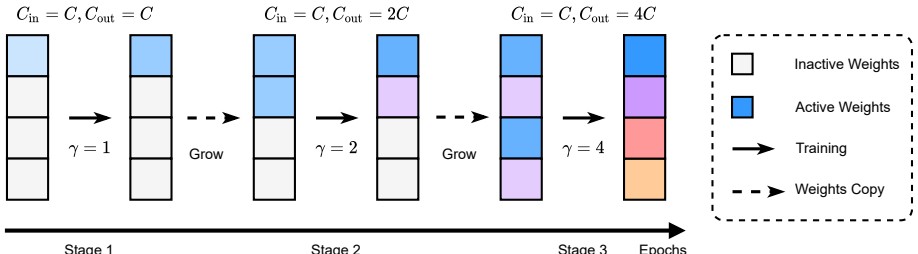

Figure 5: Illustration of our growing process of the MLP ratio $\gamma$. Taken an $\phi_1(\cdot)$ with the original $\gamma = 4$ as the example, the $4 \times 1$ matrix projects $C$ dimensions to $4C$ dimensions, with 4 rows as the output dimensions and 1 column as the input dimension. The rows are divided into $M = 4$ parts to activate progressively, from $C$, $2C$ to $4C$ at each stage. The gray cell indicates the inactive weights and the colored cells indicates the active weights. During the switch of each training stage, the weights are copied from the active parts to the inactive parts, achieving a full weight matrix.

Nevertheless, considering the computational intractability of the integration and the potential additional inefficiency when frequently switching the training cost $\mathcal{C}(T; \mathcal{M})$ at different epochs, we simplify the optimization problem by dividing the total training process into several stages and hypothesize the training cost remains unchanged in each stage. Specifically, we divide a training procedure with $T_{\max}$ epochs into $K$ stages satisfying $\sum_{k=1}^{K} T_k = T_{\max}$, and we denote $\mathcal{C}_k$ as the training cost of model $\mathcal{M}$ at the $k$-th stage. In this way, the optimization can be formulated as:

$$\min_{\mathbf{\Theta}} \mathcal{L}(f(\mathbf{X}; \mathbf{\Theta}), \mathbf{Y}), \quad \text{s.t.} \ \sum_{k=1}^{K} \mathcal{C}_k T_k \leq \mathcal{B}, \tag{6}$$

where $f(\cdot; \mathbf{\Theta})$ is the deep neural network $\mathcal{M}$ parameterized by its weights $\mathbf{\Theta}$, $\{(\mathbf{X}, \mathbf{Y}) \in \mathcal{D}\}$ is the input-label pair of training samples and $\mathcal{L}(\cdot, \cdot)$ defines the loss function of the objective. Also, it is noticeable that when we set $\mathcal{B} = \mathcal{C}_K$ and $T_{\max} = T_K = \mathcal{B}/\mathcal{C}_K$, it decreases to the common training schedule where full model is trained for all training epochs.

## 4.2 Leveraging Redundancies in Vision Transformer

Most previous works (Li et al., 2020; Chen et al., 2022a) that focus on budgeted training propose to adjust the length of training epochs to meet the specific demand on the total training cost. Despite its simplicity, these approaches usually suffer from inherent limitations that fail to make use of the model's characteristics. Nevertheless, the analysis in Sec. 3 motivates us to tackle this task from an alternative perspective, where we focus on leveraging the redundancies of Vision Transformers along the training process.

Specifically, we focus on the number of attention heads, MLP hidden dimension, and the number of visual tokens that have shown great redundancies in Vision Transformers, and propose to reduce their complexity with respect to the training stages. Given a Vision Transformer with $M$ attention heads. Denote $C$ as the MLP hidden dimension and $N$ as the number of visual tokens for each input image. At the first stage of training, only part of the attention heads are activated and the multi-head self-attention in Eq.(3) can be reformulated as

$$\mathbf{z} = \text{CONCAT}(\mathbf{h}^{(1)}, \mathbf{h}^{(2)}, \ldots, \mathbf{h}^{(M^{(1)})})\mathbf{W}_{\mathbf{O}}^{(1)}, \tag{7}$$

where $M^{(1)}$ denotes the number of activated head, and $\mathbf{W}_{\mathbf{O}}^{(1)} \in \mathbb{R}^{\frac{m}{M}C \times C}$ contains the $m/M$ part of the input dimension of the original projection matrix $\mathbf{W}_{\mathbf{O}}$. The projection matrices $\mathbf{W}_{\mathbf{q}}, \mathbf{W}_{\mathbf{k}}, \mathbf{W}_{\mathbf{v}}$ for queries, keys and values are adjusted in the same way by activating $m/M$ of their output dimensions, i.e., $\mathbf{W}_{\mathbf{q}}^{(1)}, \mathbf{W}_{\mathbf{k}}^{(1)}, \mathbf{W}_{\mathbf{v}}^{(1)} \in \mathbb{R}^{C \times \frac{m}{M}C}$. Similarly, for MLP hidden dimension, only $C_2^{(1)}$ of $C_2$ channels are activated in the MLP layers by incorporating a smaller MLP ratio $\gamma$. We illustrate the growing process of the MLP ratio in Fig. 5, and the growing of attention heads follow the same recipe.

As for the number of visual tokens, we manually drop some of the tokens, and only reserve $N^{(1)}$ to extract the image features. In practice, we set higher prior on the center region of the image, and mainly drop the tokens at the edges of image to avoid severe information loss.

As the training progresses, the redundancies in the aforementioned aspects are gradually decreased. Also, model performances will be highly restricted as the model complexity is limited. Therefore, for later stages in the training schedule, we propose to gradually turn on the inactive part of the model, and eventually recover the model capacity by activating the whole model.

However, comparing to the activated components that have been trained for a certain of epochs, the inactive part remains the same status as initialization, which is usually randomly sampled from a certain distribution. In this way, simply adding these parts to the training process may result in an imbalance of optimization, and results in degraded model performances. This problem is inevitable when activating the learnable weights for additional attention heads and MLP ratio. Therefore, to avoid the training instability, we propose to make use of the activated parts, and use their weights as the initialization. To avoid having same gradient with copied parameters, we choose to drop the statistical information of the parameters reserved in the optimizer, including their first and second order momenta in the AdamW. Consequently, this results in a diverse gradient direction, and successfully maintain the training stability.

### 4.3 ADAPTING VISION TRANSFORMER TO BUDGETS

By leveraging the technique we described in Sec. 4.2, the training cost of the models at different stages $\mathcal{C}_k$ are controlled according to the fraction of activated components. In this way, given any training budget $\mathcal{B}$, we carefully adjust the duration of each stages $T_k$ and finally satisfies the training constraint $\sum_{k=1}^{K} \mathcal{C}_k T_k \leq \mathcal{B}$ in Eq.(6).

Specifically, we employ a family of exponential functions as the prior distribution to sample the duration of each training stage. $K$ random seeds are first sampled from a uniform distribution in the range $(0, 1)$, and then mapped by an exponential function parameterized by a scalar $\alpha$: $t_k = \exp(\alpha s_k)$. The scalar $\alpha$ practically reflects the sampling bias towards different stages, *i.e.,* a larger $\alpha$ induces larger $t_k$s for the later training stages and smaller $t_k$s for the early ones. Finally, to fit the total training cost into given training budget, we linearly scale the duration of each stages:

$$T_k = \lfloor \mathcal{B} / \sum_{k=1}^{K} b_k \rfloor t_k. \tag{8}$$

## 5 EXPERIMENTS

### 5.1 EXPERIMENTAL SETUP

**Implementation details.** We follow most of the hyper-parameters of these models including model architectures, data augmentations, and stochastic depth rate (Huang et al., 2016). As discussed in Sec. 4.2, we adjust three factors controlling the training cost of the model, including the number of the activated attention heads $M$, the MLP hidden dimension $C$, and the proportion of patch tokens $N$. For all ViT models, we choose a moderate $\alpha = 2.0$ in $K = 3$ training stages, which are carefully ablated and discussed in Sec. 5.3. More detailed specifications are summarized in Appendix A.

**Datasets and models.** We mainly evaluate our method on the popular ImageNet-1K (Deng et al., 2009) dataset for large scale image recognition. We choose three famous families of ViTs, DeiT (Touvron et al., 2021), Pyramid Vision Transformer v2 (PVTv2) (Wang et al., 2022a), and Swin Transformer (Liu et al., 2021). DeiT is a classical yet powerful baseline of the isotropic ViTs while PVTv2 and Swin Transformer serve as strong baselines of the ViTs with multiscale feature structures. The transfer learning results like object detection and semantic segmentation are presented in Appendix B.

**Baselines.** In the field of budgeted training, there are two approaches by scheduling learning rate, among which Linear-LR (Li et al., 2020) propose a linear learning rate decay schedule and REX-LR (Chen et al., 2022a) design a schedule reflecting the exponential decay. We select these two popular and recent methods as our baselines, which adapt to the given training budget by compressing total epochs in training. For example, let $T_{\max}$ be the maximum epoch in training at full budget, the two baselines set the epochs to $T = 0.5T_{\max}$ to meet the 50% budget. Specifically, Linear-LR and REX-LR formulate the learning rate schedules $\eta_{\text{linear}}$ and $\eta_{\text{REX}}$ as

$$\eta_{\text{linear}}(t) = \eta_0(1 - t/T), \qquad \eta_{\text{REX}}(t) = \eta_0 \left( \frac{1 - t/T}{0.5 + 0.5(1 - t/T)} \right), \tag{9}$$

where $\eta_0$ is the initial learning rate, $t, T$ are the current epoch and the total epochs in training.

Table 1: Comparisons of our framework with the other baselines in budgeted training. The training cost measured in GFLOPs is the sum of the FLOPs of the model over all the training epochs. Linear and REX are two baselines by training full models with scheduled learning rates in the condensed epochs. The schedule displays the number of epoch in training the model at each stage.

| | **(a) DeiT-S** | | | **(b) PVTv2-b2-linear** | | | **(c) Swin-S** | | |
|---|---|---|---|---|---|---|---|---|---|
| | Schedule | Training cost | Top-1 Acc. | Schedule | Training cost | Top-1 Acc. | Schedule | Training cost | Top-1 Acc. |
| **100% training budget** | | | | | | | | | |
| original | [0,0,300] | 1382.4 | 79.8 | [0,0,300] | 1173.0 | 82.1 | [0,0,300] | 2630.7 | 83.0 |
| Ours | [86,105,243] | 1379.1 | **80.9** | [65,87,246] | 1169.1 | **82.4** | [38,124,242] | 2623.9 | **83.4** |
| **75% training budget** | | | | | | | | | |
| Linear | [0,0,225] | 1036.8 | 79.6 | [0,0,225] | 879.8 | 81.7 | [0,0,225] | 1973.0 | 82.8 |
| REX | [0,0,225] | 1036.8 | 79.6 | [0,0,225] | 879.8 | 81.8 | [0,0,225] | 1973.0 | 82.7 |
| Cosine | [0,0,225] | 1036.8 | 79.4 | [0,0,225] | 879.8 | 82.0 | [0,0,225] | 1973.0 | 82.9 |
| Ours | [55,55,193] | 1032.1 | **80.1** | [51,118,160] | 877.1 | **82.2** | [42,123,167] | 1967.9 | **83.0** |
| **50% training budget** | | | | | | | | | |
| Linear | [0,0,150] | 691.2 | 78.0 | [0,0,150] | 586.5 | 81.4 | [0,0,150] | 1315.4 | 82.1 |
| REX | [0,0,150] | 691.2 | 78.1 | [0,0,150] | 586.5 | 81.3 | [0,0,150] | 1315.4 | 82.0 |
| Cosine | [0,0,150] | 691.2 | 77.9 | [0,0,150] | 586.5 | 81.3 | [0,0,150] | 1315.4 | 82.0 |
| Ours | [49,71,113] | 689.8 | **78.9** | [25,27,132] | 584.3 | **81.8** | [41,42,126] | 1311.9 | **82.4** |
| **25% training budget** | | | | | | | | | |
| Linear | [0,0,75] | 345.6 | 73.1 | [0,0,75] | 293.3 | 79.3 | [0,0,75] | 657.7 | 79.5 |
| REX | [0,0,75] | 345.6 | 73.2 | [0,0,75] | 293.3 | 79.3 | [0,0,75] | 657.7 | 79.7 |
| Cosine | [0,0,75] | 345.6 | 72.7 | [0,0,75] | 293.3 | 78.9 | [0,0,75] | 657.7 | 79.1 |
| Ours | [29,49,50] | 343.7 | **74.5** | [27,28,56] | 290.5 | **79.6** | [29,35,55] | 648.0 | **80.0** |

## 5.2 BUDGETED TRAINING OF VITS

We report the results of our method and the budgeted training baselines on DeiT-S (Touvron et al., 2021), PVTv2-b2linear (Wang et al., 2022a), and Swin-S (Liu et al., 2021). The training cost is the sum of the FLOPs of the model along all the epochs in training. We choose 25%, 50% and 75% for budgeted training. For the learning rate scheduler baselines, we directly discount the number of epochs from 300 to 75, 150, and 225, respectively. For our method, we sample each $T_k$ under the given budget to fit the constraint by Eq.(8). We summarize our results in Tab. 1, in which our method outperforms two baselines consistently on the three models in terms of Top-1 accuracy on ImageNet-1K validation dataset. For DeiT-S, our method achieves +1.3%, +0.8%, and +0.5% over the baselines on the three training budgets. For The results on PVTv2 and Swin-S models also shows the superiority of our method to the learning rate scheduling techniques. When using full training budget, our method has significant improvements in 1.0% over the original models. Fig. 7 in Appendix also illustrates the effectiveness of our method. More comparisons with GradMatch (Killamsetty et al., 2021) which reduces the training costs by selecting valuable data, are reported in Appendix C.

## 5.3 ANALYSES & ABLATION STUDY

**Restricting training epochs.** We add another extra constraint that the number of total epochs is limited to 300 as the original model to check the flexibility of our framework. Tab. 2 shows that if the total epochs remain unchanged as the training budget goes low, the training cost is likely to exceed the budget and the performances of models are faded. Therefore, the total epochs should be adjusted according to the budgets in the scheme of multiple stages of different model complexities.

**Training time.** We report the training time of our methods on DeiT-S in Tab. 3. Under 75% FLOPs training budget, our framework saves the training time over 11% in terms of GPU hours without performance drop. Under other budgets, our method also achieves considerable time saving with competitive model performances. Because the total training time is heavily influenced by the CPUs and I/O of the file system, we measure the time of forward and backward passes only to further assess the time saving. It is observed that the saved foward and backward time is more close to the saved training budgets, which implies our method could save more training time in total provided optimized hardware systems.

Table 2: Results of training DeiT-S with 300 epoch constraints under different budgets. Training cost is in GFLOPs.

| Budget | Schedule | Training cost | Top-1 Acc. |
|---|---|---|---|
| 100% | [0,100,200] | 1112.1 | 80.1 |
| 75% | [25,100,175] | 1014.2 | 79.6 |
| 50% | [100,100,100] | 720.3 | 78.6 |
| 25% | [225,50,25] | 365.7 | 72.2 |

Table 3: Training time of our method on DeiT-S. All the records of the time are measured on 8 RTX 3090 GPUs. FP&BP is the span of the forward and backward passes.

| Model | Budget | Total GPU hours | FP&BP hours | Top-1 Acc. |
|---|---|---|---|---|
| DeiT-S original | - | 187 | 65 | 79.8 |
| DeiT-S [55,55,193] | 75% | 168 | 52 | 80.1 |
| DeiT-S [49,71,113] | 50% | 124 | 34 | 78.9 |
| DeiT-S [22,26,60] | 25% | 59 | 17 | 74.4 |

Table 4: Ablation on the choices of $\alpha$ in sampling epochs for different stages of training DeiT-S. Schedules are all sampled under about 25% training budget. Training cost is in GFLOPs.

Table 5: Results on various types of schedule functions. max $t_2$ denotes the schedule with the longest epoch in the second stage of training DeiT-S. Schedules are all sampled under the 25% training budget. Training cost is measured in GFLOPs.

Figure 6: The accuracies of DeiT-S trained under different number of stages $K$ with similar training costs are plotted.

| $\alpha$ | Schedule | Training cost | Top-1 Acc. |
|---|---|---|---|
| +2.0 | [29,49,50] | 343.7 | 74.5 |
| +5.0 | [1,16,67] | 339.9 | 74.4 |
| 0.0 | [47,47,47] | 338.5 | 74.2 |
| -2.0 | [86,63,35] | 340.6 | 73.8 |
| -5.0 | [122,51,35] | 342.6 | 73.6 |

| Func. Type | Schedule | Training cost | Top-1 Acc. |
|---|---|---|---|
| $t_k = s_k$ | [31, 42, 52] | 341.0 | 74.1 |
| $t_k = 1 - s_k$ | [86, 63, 35] | 343.0 | 73.6 |
| max $t_2$ | [54, 73, 36] | 342.2 | 73.9 |

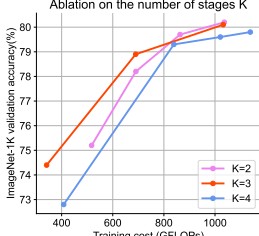

**Scalar $\alpha$ in $T_k$ sampling** controls the growth rate of the exponential function. As shown in Tab. 4, $\alpha = 5.0$ compress the first two stages in less than 20 epochs. When $\alpha$ is large enough, the epochs of last training stage $T_K$ will dominate all the training epochs, which comes to the case $\mathcal{C}_K = \mathcal{B}$ as Li et al. (2020). If $\alpha$ is set to 0, all $t_k$s are equal. And when $\alpha$ is negative, early $t_k$s will become large rather than the later $t_k$s, which results in degradation in model performance by -0.7% and -0.9%.

**Linear schedule functions** with the form $t_k = \beta s_k$ and $t_k = \beta(1 - s_k)$ are also evaluated. Since the cost of each stage is normalized to meet the total training budget (Eq. (8)), we simply set $\beta = 1$ without loss of generality. As shown in Tab. 5, linear functions also show the trend that increasing functions works well while decreasing functions shade the performances. The longest epoch for the second training stage whose result is displayed in the last row, achieves a moderate performance.

**The number of stage $K$.** To verify the effectiveness of different stage number $K$ in our proposed framework, we choose $K = 2, 3, 4$ and evaluate the schedules under the similar budgets in the paper. The results are illustrated in Fig. 6, from which we find that $K = 3$ outperforms $K = 2, 4$ in all the budgets, thus $K = 3$ is adopted in our method. More ablation results are reported in Appendix D.

## 6 CONCLUSION

This paper presents a novel framework for training Vision Transformers at *any given budget* by reducing the inherent redundancies of the model at the early training stages. We investigate three redundancy factors in model structure, including attention heads, hidden dimensions in MLP, and visual tokens. Based on these observations, we propose a training strategy to dynamically adjust the activation rate of the model along the training process. Extensive experiments show the effectiveness of our framework with competitive performances on a wide range of training budgets.

## ACKNOWLEDGMENTS

This work is supported in part by National Key R&D Program of China (2021ZD0140407), the National Natural Science Foundation of China (62022048, 62276150) and THU-Bosch JCML.

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

# APPENDIX

## A  IMPLEMENTATION DETAILS

We list our configurations of the models in the experiments in Table 6. We present the detailed specification of DeiT-S, DeiT-B (Touvron et al., 2021), PVTv2-b2-linear (Wang et al., 2022a) and Swin-S (Liu et al., 2021), including their numbers of activated attention heads $M$, activated hidden dimensions in the MLP blocks $C$, and activated patch tokens $N$. Note that PVTv2-b2-linear and Swin-S has 4 model stages, we modify these factors at each stage separately in 4 columns, respectively. Generally, we let $M$ and $N$ grow linearly, e.g., $M$ in 2, 4, 6, and $N$ in 50%, 75% and 100%. And for the hidden dimensions, we set the MLP ratio $\gamma$ to grow in 1, 2, 4. As for the number of epochs at each training stage. To reduce $N$, we simply perform center cropping on the patch tokens.

The approach can generalize to Swin Transformer (Liu et al., 2021) by adopting different sizes of the windows at different training stage. For example, when the spatial sizes of the tokens are $40 \times 40, 48 \times 48, 56 \times 56$ at the three training stages, the window sizes for window attention are $5 \times 5, 6 \times 6, 7 \times 7$, respectively. As for the head numbers and the MLP ratio, they are set just following the ones in the DeiT models. The results of DeiT-B are reported in Appendix E.

Table 6: Configurations of the models evaluated in the budgeted training experiments.

| Model | | DeiT-S | DeiT-B | PVTv2-b2-linear | | | | Swin-S | | | |
|---|---|---|---|---|---|---|---|---|---|---|---|
| Training Stage 1 | $M$ | 2 | 4 | 1 | 1 | 2 | 2 | 1 | 2 | 4 | 8 |
| | $C$ | 384 | 768 | 256 | 512 | 320 | 512 | 96 | 192 | 384 | 768 |
| | $N$ | 100 | 100 | 1600 | 400 | 100 | 25 | 1600 | 400 | 100 | 25 |
| | FLOPs | 0.69G | 2.57G | 0.80G | | | | 1.32G | | | |
| Training Stage 2 | $M$ | 4 | 8 | 1 | 2 | 4 | 4 | 2 | 4 | 8 | 16 |
| | $C$ | 768 | 1536 | 512 | 1024 | 640 | 1024 | 192 | 384 | 768 | 1536 |
| | $N$ | 144 | 144 | 2304 | 576 | 144 | 36 | 2304 | 576 | 144 | 36 |
| | FLOPs | 1.91G | 7.23G | 1.79G | | | | 3.64G | | | |
| Training Stage 3 | $M$ | 6 | 12 | 1 | 2 | 5 | 8 | 3 | 6 | 12 | 24 |
| | $C$ | 1536 | 3072 | 1024 | 2048 | 1280 | 4096 | 384 | 768 | 1536 | 3072 |
| | $N$ | 196 | 196 | 3136 | 784 | 196 | 49 | 3136 | 784 | 196 | 49 |
| | FLOPs | 4.61G | 17.58G | 3.91G | | | | 8.77G | | | |

## B  TRANSFER LEARNING EVALUATION

To verify the performances of the pre-trained models under budgeted training, we evaluate these models on several downstream benchmarks, including CIFAR-10/100 (Krizhevsky et al., 2009) transfer learning, MS-COCO (Lin et al., 2014) object detection and instance segmentation, and ADE20K (Zhou et al., 2017) semantic segmentation. It is observed that the models trained in our framework with 75% budgets achieve similar ImageNet-1K classification accuracy to the ones in the original training scheme. Therefore, we choose the models trained under 75% budgets to evaluate on these downstream tasks for convenient comparisons.

**Transfer learning on smaller datasets.** For CIFAR-10/100 transfer learning task, we follow the procedure in the official DeiT repository[1] to finetune the pre-trained DeiT-S (Touvron et al., 2021) models. The results are reported in Tab. 7, in which our method under 75% training budgets slightly outperforms the original model on both CIFAR-10 and CIFAR-100 datasets.

**Object detection and instance segmentation.** For MS-COCO object detection and instance segmentation task, we evaluate Mask R-CNN (He et al., 2017) with PVTv2-b2-linear (Wang et al., 2022a) backbone in 1x schedule and Swin-S (Liu et al., 2021) model in 3x schedule. The schedules 1x & 3x in object detection tasks mean decaying learning rate by 0.1 after the 8[th], 11[st] epoch in 12 total epochs and the 27[th], 33[rd] epoch in 36 total epochs, respectively. We use the popular object detection benchmark codebase, MMDetection (Chen et al., 2019), for fair comparisons. As shown in Tab. 9, our method under 75% training budgets achieves comparable performances on both models.

---

[1]https://github.com/facebookresearch/deit/issues/45

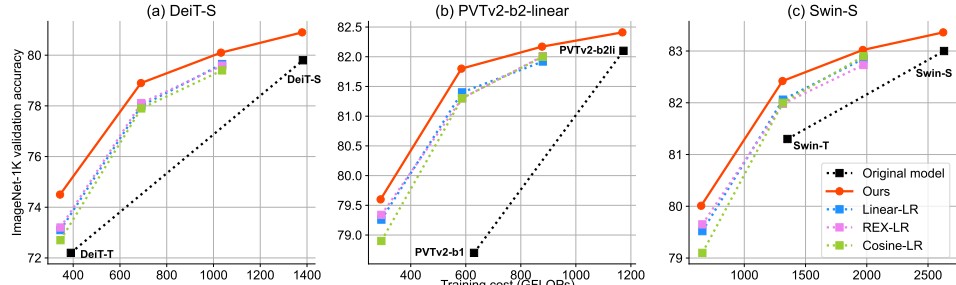

Figure 7: The budgeted training results of our methods and other baselines. (a)(b)(c) plot the accuracy under different training cost of DeiT-S, PVTv2-b2-linear and Swin-S models, where our method outperforms the baselines under the same training budget.

Table 7: Transfer learning results of DeiT-S models on CIFAR-10 and CIFAR-100.

| Pretrain method | CIFAR-10 Top-1 Acc. | CIFAR-100 Top-1 Acc. |
|---|---|---|
| Original | 98.78% | 89.44% |
| Ours (75% budget) | 98.91% | 89.65% |

Table 8: Semantic segmentation results on ADE20K.

| Pretrain method | Schedule | Segmentor | mIoU |
|---|---|---|---|
| PVTv2-b2linear | 40K | S-FPN | 45.10 |
| Ours (75% budget) | 40K | S-FPN | 45.29 |
| Swin-S | 160K | UperNet | 47.64 |
| Ours (75% budget) | 160K | UperNet | 47.46 |

Table 9: Object detection and instance segmentation results of Mask R-CNN on MS-COCO.

| Pretrain method | Schedule | $AP^b$ | $AP^b_{50}$ | $AP^b_{75}$ | $AP^b_s$ | $AP^b_m$ | $AP^b_l$ | $AP^m$ | $AP^m_{50}$ | $AP^m_{75}$ | $AP^m_s$ | $AP^m_m$ | $AP^m_l$ |
|---|---|---|---|---|---|---|---|---|---|---|---|---|---|
| PVTv2-b2linear | 1x | 44.1 | 66.3 | 48.4 | 28.0 | 47.4 | 58.0 | 40.5 | 63.2 | 43.6 | 21.5 | 43.0 | 58.2 |
| Ours (75% budget) | 1x | 44.1 | 66.1 | 48.2 | 28.3 | 47.4 | 57.1 | 40.3 | 63.3 | 43.0 | 24.7 | 43.5 | 54.2 |
| Swin-S | 3x | 48.5 | 70.2 | 53.5 | 33.4 | 52.1 | 63.3 | 43.3 | 67.3 | 46.6 | 28.1 | 46.7 | 58.6 |
| Ours (75% budget) | 3x | 48.2 | 70.2 | 53.1 | 32.1 | 51.7 | 62.6 | 43.2 | 67.0 | 46.6 | 27.3 | 46.8 | 58.3 |

**Semantic segmentation.** For ADE20K semantic segmentation task, we follow PVT-v2 (Wang et al., 2022a) settings to evaluate our method on Semantic FPN (Kirillov et al., 2019) for 40K training steps and follow Swin Transformer (Liu et al., 2021) to apply the backbone on UperNet (Xiao et al., 2018) for 160K steps. We use MMSegmentation (Contributors, 2020) to perform the experiments. Tab. 8 that demonstrates our method is competitive to the original models.

## C    COMPARISON WITH DATASET PRUNING

For dataset pruning or coreset selection approaches, we choose a recent work, GradMatch (Killamsetty et al., 2021), as our baseline. GradMatch leverages the orthogonal matching pursuit algorithm to match the gradients of the training and validation set, for an adaptive seletion of subsets. On ImageNet dataset, GradMatch provides the dataset pruning results on 5%, 10% and 30% budgets of ResNet-18 (He et al., 2016). In GradMatch experiments, ResNet-18 with 11.7M parameters and 1.82GFLOPs is trained in 350 epochs with coreset selection at every 20 epochs, thus consumes the training cost of 31.85G, 63.70G, and 191.1G FLOPs under three budgets respectively. To match this training cost and model size, we adopt DeiT-T (Touvron et al., 2021), which has 5.7M parameters and 1.26GFLOPs to perform budgeted training. For fair comparison, we choose the GradMatch variant without a warm up that uses full dataset to pretrain the model for a better initial data pruning and we choose the per-batch version denoted by GradMatch-PB for its improved performance. Tab. 10 shows our method significantly outperforms GradMatch under all three training budgets on ImageNet-1K dataset.

## D    MORE ABLATION STUDY

**Different activated components.** As summarized in Tab. 13, we evaluate different types of combination of the activated components on DeiT-S with a fixed schedule [50,100,150]. $M$, $N$, $C$

Table 10: Comparison to the dataset pruning method. The results of GradMatch (Killamsetty et al., 2021) are excerpted from Tab.5 in the paper. Training cost is measured in GFLOPs.

| Method | Model | Schedule / Fraction | Training cost | Top-1 Acc. |
|---|---|---|---|---|
| GradMatch-PB | ResNet-18 | 5% of ImageNet | 31.9G | 45.15 |
| Ours | DeiT-T | [11,15,17] | 31.4G | **57.88** |
| GradMatch-PB | ResNet-18 | 10% of ImageNet | 63.70G | 59.04 |
| Ours | DeiT-T | [8,24,39] | 63.23G | **60.20** |
| GradMatch-PB | ResNet-18 | 30% of ImageNet | 191.10G | 68.12 |
| Ours | DeiT-T | [22,51,127] | 190.85G | **69.49** |

Table 11: Ablation of different ways to growing the linear projections in the [75, 100, 125] schedule of the DeiT-S model.

| params | states | $\times 0.5$ | +noise | Top-1 Acc. |
|---|---|---|---|---|
| ✗ | ✗ | ✗ | ✗ | 75.8 |
| ✓ | ✗ | ✗ | ✗ | 79.4 |
| ✓ | ✗ | ✓ | ✗ | 79.3 |
| ✓ | ✓ | ✗ | ✗ | 76.5 |
| ✓ | ✓ | ✓ | ✗ | 78.5 |
| ✓ | ✓ | ✓ | ✓ | 78.5 |

Table 12: Baselines of using fewer patch tokens by downsampling the input images. The input resolution is scaled down from $224^2$ to $192^2$, $160^2$, and $112^2$ to reduce the number of tokens.

| Epoch | #tokens | Avg.FLOPs | Top-1 Acc. |
|---|---|---|---|
| 300 | 196 (100%) | 4.6G | 79.8 |
| 300 | 144 (75%) | 3.3G | 78.5 |
| 300 | 100 (50%) | 2.3G | 75.7 |
| 300 | 49 (25%) | 1.1G | 66.2 |

mean adopting the attention heads, tokens and MLP hidden dimensions as the activated components, respectively. The more components partially activated we used in budgeted training, the more training cost we save. We also observe that the number of tokens $N$ is the key to reducing training cost.

**Using fewer tokens.** We provide another baseline of using fewer tokens by selecting about 25%, 50% and 75% patch tokens in DeiT-S, as shown in Tab. 12. We observe that only reducing the tokens to save space consumption leads to a sharp performance drop.

**The implementation of the weights copy.** Different ways to growing the weight matrices of the MHSA layers and the MLP layers are listed in Tab. 11. The random initialization for the newly activated dimensions results in a poor performance, as listed in the first row. The second row shows that directly copying parameters brings about a good performance. Nevertheless, copying the optimizer states along with the parameters is damaging for the symmetry concern. The compensation factor 0.5 of the replications proposed in (Chen et al., 2015) helps to lift the accuracy to 78.5. However only using this $\times 0.5$ technique or adding noise to the grown weights do not help a lot.

Table 13: Ablation on different activated model components controlling complexity with schedule [50,100,150].

| DeiT-S | Training cost | Top-1 Acc. |
|---|---|---|
| $M+N+C$ | 916.2 | 79.5 |
| $M+N$ | 985.9 | 79.9 |
| $M+C$ | 1021.7 | 80.0 |
| $N+C$ | 1074.9 | 80.0 |
| $M$ only | 1265.6 | 80.2 |
| $N$ only | 973.9 | 80.1 |
| $C$ only | 1138.5 | 80.2 |

# E RESULTS ON DEIT-B

For DeiT-B, we evaluate our method under the same budgets of 25%, 50% and 75%. As shown in Tab. 14, there are almost no performance drops of our method as the budgets go smaller whereas the baselines degrade in a large magnitude. Because DeiT-B is a much wider model than DeiT-S, it has more redundancies that can be exploited during training. Nonetheless, larger models require more iterations to converge than the smaller ones, which explains the poor results of the baselines.

Table 14: Comparison of our methods and other baselines on DeiT-B. Training cost is measured in GFLOPs.

| DeiT-B | Schedule | Cost | Top-1 Acc. |
|---|---|---|---|
| original | [0,0,300] | 5274.9 | 81.8 |
| Linear | [0,0,225] | 3956.2 | 79.6 |
| REX | [0,0,225] | 3956.2 | 79.1 |
| Ours | [50,50,200] | 3702.4 | **81.2** |
| Linear | [0 0,150] | 2637.5 | 78.6 |
| REX | [0,0,150] | 2637.5 | 79.2 |
| Ours | [100,100,100] | 2122.7 | **81.1** |
| Linear | [0,0,75] | 1318.7 | 77.3 |
| REX | [0,0,75] | 1318.7 | 77.4 |
| Ours | [150, 100, 50] | 1372.2 | **80.9** |

