# OpenReview forum: "Budgeted Training for Vision Transformer"
_ICLR.cc/2023/Conference — ICLR 2023 poster_

### Official Review · Reviewer_jJMJ · 2022-10-23

**Confidence:** 4
**Correctness:** 3
**Technical Novelty And Significance:** 2
**Empirical Novelty And Significance:** 3
**Recommendation:** 6

**Clarity, Quality, Novelty And Reproducibility:**

The paper is easy to read and understand. Making training more efficient is an interesting research area and can have a lot of impact. The proposed method seems novel and shows good results for several architectures and training budgets. There are some implementation details in the appendix so it should not be too difficult to reproduce some experiments.

**Strength And Weaknesses:**

**Strengths**
- To reduce the training cost of Vision Transformer, the paper identifies some redundancies in the architectures. Section 3.2, 3.3 and 3.4 analyzes the redundancies in the attention heads, MLPs and visual tokens of Vision Transformer architectures. For each part, an empirical analysis performed on a subset of ImageNet is shown. The analyses show that the redundancies are usually high at the beginning of the training and then gradually decrease.
- After identifying some redundancies, the paper introduces a 3 phase training strategy that takes advantage of the redundancies. The training starts with a small architecture and the architecture grows after each phase because small architectures can be trained faster than large architectures and large architectures have more redundancies.
- I like the research area on budgeted training. I think that making from scratch training more accessible/affordable is positive for the community because more research labs will be able to develop new model architectures. I think it can also help to reduce the environmental impact of ML model training.
- The proposed method is evaluated on ImageNet with multiple Vision Transformer architectures (DeiT, Pyramid Vision Transformer v2 (PVTv2), and Swin Transformer). The proposed training strategy improves the performances for several training budgets (25%, 50%, 75%, and 100%). The paper also contains ablation studies of the important hyper-parameters of the proposed method.


**Weaknesses**
- The performances are evaluated only on ImageNet. It could be interesting to evaluate the pre-trained models on downstream tasks to analyze the transfer learning performances because it will increase the quality of the paper.
- The exponential function used to define the training budget for each phase does not seem well justified.
- The proposed training strategy has 3 phases but there is no justification about this design choice. It could be interesting to show results with more phases.



**Summary Of The Paper:**

This paper introduces a method to train a Vision Transformer given a budget defined by a total training time or computation cost. Training a Vision Transformer from scratch is usually expensive, so designing a budgeted training is a way to make the training a Vision Transformer more accessible. The paper analyzes and identifies some redundancies in the attention heads, MLPs and visual tokens of Vision Transformer architectures. The redundancies are usually high at the beginning of the training and then gradually decrease. To make training faster, the paper introduces a 3 phase training strategy, where the training starts with a small architecture and the architecture grows after each phase. The proposed method is evaluated on ImageNet with multiple Vision Transformer architectures.

**Summary Of The Review:**

Overall, I think the paper is well motivated and is about an interesting research area. The proposed method seems novel and improves the performances in multiple scenarios but the experiment section could be improved.

---

> ### Author Response · Authors · 2022-11-18
> **Response to Reviewer jJMJ**
>
> We appreciate the reviewer's valuable comments. Please find our response to your concerns in the following：
>
> - **Transfer Learning.** We add more experiment results of transferring the pre-trained models by our method to the downstream tasks, including classification finetuning, object detection & instance segmentation, and semantic segmentation. On these tasks, we evaluate the models trained under a 75% budget whose classification accuracies on ImageNet-1K are similar to the original models. We apply DeiT-S on CIFAR-10 and CIFAR-100 image classification as the direct application to dense prediction tasks is nontrivial. For multiple-scale architectures, we employ PvTv2-b2linear and Swin-S on MS-COCO object detection and instance segmentation with Mask R-CNN, and ADE20K semantic segmentation with Semantic FPN and UperNet. The results are briefly summarized in the tables below. These experiment settings are identical to those in their original paper, more detailed configurations and results are filed in Sec. B of the appendix.
>
> Image classification | CIFAR-10 | CIFAR-100
> :-- | :--: | :--:
> DeiT-S (Original) | 98.78% | 89.44%
> Ours (75% budget) | 98.91% | 89.65%
>
> Obejct detection & Instance segmentation | Detector | Schedule | Box AP | Mask AP
> :-- | :--: | :--: | :--: | :--:
> PVTv2-b2-linear (Original) | MaskRCNN | 1x | 44.1 | 40.5
> Ours (75% budget) | MaskRCNN | 1x | 44.1 |40.3
> Swin-S (Original) | MaskRCNN | 3x | 48.5 | 43.3
> Ours (75% budget) | MaskRCNN | 3x | 48.2 | 43.2
>
> Semantic Segmentation | Segmentor | Schedule | mIoU
> :-- | :--: | :--: | :--:
> PVTv2-b2-linear (Original) | Semantic FPN | 40K | 45.10
> Ours (75% budget) | Semantic FPN | 40K | 45.29
> Swin-S (Original) | UperNet | 160K | 47.64
> Ours (75% budget) | UperNet | 160K | 47.46
>
> - **Exponential function.** Honestly, there may exist some sophisticated approaches to better determine the epochs of each training phase, such as some search-based method. However, we find in fact that the  exact number of epochs or the form of scheduling function does not account for so much. The family of exponential functions is well capable of modeling a wide range of distributions of the schedules and is suitable for various problems. Additionally, adjusting the rate factor $\alpha$ can generate flexible portions of epochs in each training stage. We provide some other choices of scheduling functions in Sec. C of the appendix, with the increasing and decreasing linear functions and setting the maximum epochs to the second stage. The other choices do not outperform the exponential functions. In fact, from Tab.4 and Tab.10, it is the key that increasing monotonicity can achieve good performances. Therefore, an increasing exponential function with $\alpha=2$ is adapted to meet the demand for flexibility and monotonicity.
>
>
> Budget | Function Type | Schedule | Cost | Top-1 Acc.
> :--: | :--: | :--: | :--: | :--:
> 25% | $t_k=s_k$ | [31, 42, 52] | 341.0G | 74.1
> 25% | $t_k=1-s_k$ | [86, 63, 35] | 343.0G | 73.6
> 25% | max $t_2$ | [54, 73, 36] | 342.2G | 73.9
>
>
> - **The number of training phases.** We have investigated this factor in the ablation studies of the paper. As discussed in Sec5.4, we choose K=2, 3, 4 to verify its influence. From the results in Fig.7, we find that K = 3 outperforms K = 2, 4 under all the budgets. On the one hand, introducing too many training phases would cause lots of switches between stages with fewer epochs in each stage. However, the model redundancies are limited, therefore inadequate training in the early stages may result in suboptimal convergence. On the other hand, reducing training phases would hedge the flexibility of the schedules and struggle with the dilemma between the budgets and the model performances.
>
> We kindly ask the reviewer to reassess the paper in light of this comment if it clears things up. We are happy to answer more if you have any remaining concerns or questions.

---

> ### Author Response · Authors · 2022-11-27
> **A gentle reminder to Reviewer jJMJ**
>
> Dear Reviewer **jJMJ**,
>
> We thank you again for your insightful comments and your time reviewing our paper.
>
> We would appreciate it if you could confirm that our responses address your concerns. We are happy to answer more if you have any remaining concerns or questions.
>
> Best regards,
>
> Authors

---

> ### Author Response · Authors · 2022-12-08
> **A kindly reminder to Reviewer jJMJ**
>
> Dear Reviewer **jJMJ**,
>
> Thank you again for your precious time!
>
> We kindly ask the reviewer to reassess the paper in light of this comment if it clears things up. We are happy to answer more if you have any remaining concerns or questions.

---

### Official Review · Reviewer_vaAP · 2022-10-25

**Confidence:** 3
**Correctness:** 3
**Technical Novelty And Significance:** 2
**Empirical Novelty And Significance:** 1
**Recommendation:** 5

**Clarity, Quality, Novelty And Reproducibility:**

This paper is clear written, All the experiments are well described and should be straightforward to reproduce.

**Strength And Weaknesses:**

Strength:
1. the analysis of the redundancy in modules of  VT will be very important not only restricted to this work, but potentially many research on designing efficient VT models.
2. The paper is quite easy to follow and all the contents are easy to follow (despite a few typos: Sec 6, effusiveness -> effectiveness?)

Weakness:
1. the motivation could be improved. There are multiple solutions for reducing training cost, they could be but not limited to data set pruning, using smaller models, or more effective training schemes. Using training cost as motivation seems not enough to justify this paper. And this will make readers wonder how this proposed method compared to other treatments. Perhaps one scenario unique is continuous/online training?
2. more discussions might be needed to convince the effectiveness of the proposed training strategy: 1. how it compares to different original training schedules, e.g., what if we reduce the original training epochs from 300 to 150, using some quick schedule (cosine learning rate)?

**Summary Of The Paper:**

This paper presents training strategy that allows to training optimal transformers under user-defined budgets.  The budget training strategy is based on investigating the redundancy in attention heads, hidden dimensions in MLP, and
visual tokens. The training strategy could adjust the activation rate of the model along the training process to make use of the redudancies. Extensive experiments have demonstrated effectiveness.

**Summary Of The Review:**

See strength/weakness

---

> ### Author Response · Authors · 2022-11-18
> **Response to Reviewer vaAP**
>
> We appreciate the reviewer's valuable comments. Please find our response to your concerns in the following：
>
> - **Motivation**. We agree with the reviewer that there exist many solutions for reducing the training cost of the model. These solutions consist of smaller model variants, dataset selection, using advanced optimizers, and efficient training schemes, as you comment. In our work, we would like to investigate this problem from the perspective of model redundancies, which has not yet been explored comprehensively. To clarify our motivation, we will revise our paper to narrow it down to focus on the point of reducing model redundancies during different phases of training. The abstract, introduction, and conclusion sections will be polished to restrict our work to the scope of model structures and redundancies.
>
> - **Other treatments.** It is a good comment for pointing out the other treatments. Our work makes a contribution to the budgeted training problem from the perspective of model redundancy. Our proposed approach compares favorably to other treatments. From Tab.1 and Fig.6 of the paper, our framework achieves better performances than using smaller models, e.g., using DeiT-T as the smaller variant of DeiT-S, and outperforms different effective training schemes including linear-LR, REX-LR, and cosine-LR schedules. In terms of the advanced optimization technique, popular existing training procedures of Vision Transformers almost adopt AdamW optimizer with cosine learning rate scheduler as best practice. To compare with the dataset pruning or coreset selection, we choose a recent approach named GradMatch [1] as our baseline. We focus on the budgeted training on ImageNet dataset and choose three budgets according to GradMatch. Since GradMatch uses ResNet-18(11.7M, 1.82G) to perform dataset pruning, we adopt DeiT-T(5.7M, 1.26G) to train under similar budgets. From the table below, our framework achieves competitive results to GradMatch under all three training budgets. More detailed experiment results are included in Tab.11 of the appendix. Moreover, our method can offer a possible combination of the dataset pruning methods to achieve more efficient budgeted training, as a future direction.
>
>
> Method | Model | Schedule / Fraction | Training cost | Top-1 Acc.
> :-- | :--: | :--: | :--: | :--:
> GradMatch-PB | ResNet-18 | 5% of ImageNet | 31.90G | 45.15
> Ours | DeiT-T | [11,15,17] | 31.42G | **57.88**
> GradMatch-PB | ResNet-18 | 10% of ImageNet | 63.70G | 59.04
> Ours | DeiT-T | [8,24,39] | 63.23G | **60.20**
> GradMatch-PB | ResNet-18 | 30% of ImageNet | 191.10G | 68.12
> Ours | DeiT-T | [22,51,127] | 190.85G | **69.49**
>
>
> - **Different original schedules.** We thank the reviewer for the valuable suggestion. We have updated the results using the original training schedule in the table below, i.e., the cosine learning rate decay rule. From these results, we observe that the cosine learning rate works well in relatively higher training budgets (75%), which outperforms or achieves competitive performance to Linear and REX learning rate schedules. However, the cosine learning rate schedule results in drastic performance degradation under smaller training budgets (25%, 50%). These results are also updated in Tab.1 and Fig.6 of the paper.
>
>
> &nbsp;| &nbsp;| &nbsp; | &nbsp; | &nbsp; | &nbsp; | Model | &nbsp; | &nbsp; | &nbsp; | &nbsp;
> :--: | :--: | :--: | :--: | :--: | :--: | :--: | :--: | :--: | :--: | :--:
> &nbsp; | &nbsp; | &nbsp; | **DeiT-S** | &nbsp; | &nbsp; | **PVTv2-b2-linear** | &nbsp; | &nbsp; | **Swin-S** | &nbsp;
> **Method** | **Budget** | **Schedule** | **Cost** | **Top-1 Acc.** | **Schedule** | **Cost** | **Top-1 Acc.** | **Schedule** | **Cost** | **Top-1 Acc.**
> cosine | 75% | [0,0,225] | 1036.8G | 79.4 | [0,0,225] | 879.8 | 82.0 | [0,0,225] | 1973.0 | 82.9
> ours | 75% | [55,55,193] | 1032.1G | **80.1** | [51,118,160] | 877.1 | **82.2** | [42,123,167] | 1967.9 | **83.0**
> cosine | 50% | [0,0,150] | 691.2G | 77.9 | [0,0,150] | 586.5 | 81.3 | [0,0,150] | 1315.4 | 82.0
> ours | 50% | [49,71,113] | 689.8G | **78.9** | [25,27,132] | 584.3 | **81.8** | [41,42,126] | 1311.9 | **82.4**
> cosine | 25% | [0,0,75] | 345.6G | 72.7 | [0,0,75] | 293.3 | 78.9 | [0,0,75] | 657.7 | 79.1
> ours | 25% | [29,49,50] | 343.7G | **74.5** | [25,27,132] | 290.5 | **79.6** | [27,28,56] | 648.0 | **80.0**
>
>
> We kindly ask the reviewer to reassess the paper in light of this comment if it clears things up. We are happy to answer more if you have any remaining concerns or questions.
>
>
> [1] Killamsetty, Krishnateja, et al. "Grad-match: Gradient matching based data subset selection for efficient deep model training." International Conference on Machine Learning. PMLR, 2021.

---

> ### Author Response · Authors · 2022-11-27
> **A gentle reminder to Reviewer vaAP**
>
> Dear Reviewer **vaAP**,
>
> We thank you again for your insightful comments and your time reviewing our paper.
>
> We would appreciate it if you could confirm that our responses address your concerns. We are happy to answer more if you have any remaining concerns or questions.
>
> Best regards,
>
> Authors

---

> ### Author Response · Authors · 2022-12-08
> **A kindly reminder to Reviewer vaAP**
>
> Dear Reviewer **vaAP**,
>
> Thank you again for your precious time!
>
> We kindly ask the reviewer to reassess the paper in light of this comment if it clears things up. We are happy to answer more if you have any remaining concerns or questions.

---

### Official Review · Reviewer_H7vm · 2022-10-28

**Confidence:** 3
**Clarity, Quality, Novelty And Reproducibility:** I have no other concers.
**Correctness:** 4
**Technical Novelty And Significance:** 4
**Empirical Novelty And Significance:** 4
**Recommendation:** 6

**Strength And Weaknesses:**

This paper investigates three redundancy factors in Vision Transformers, including attention heads, hidden dimensions in MLP, and visual tokens.

Extensive experiments show the effusiveness of the proposed framework with competitive performances on a wide range of training budgets.

**Summary Of The Paper:**

This paper presents a framework for training Vision Transformers at any given budget by reducing the inherent redundancies of the model at the early training stages.

This paper proposes a training strategy to dynamically adjust the activation rate of the model along the training process.


**Summary Of The Review:**

This paper take a step forward and focus on the problem of budgeted training and aims to achieve the highest model performance under any given training budget.

---

> ### Author Response · Authors · 2022-11-18
> **Response to Reviewer H7vm**
>
> We appreciate the reviewer's valuable comments. Please do not hesitate to post new comments if you have any other concerns.

---

> ### Author Response · Authors · 2022-11-27
> **A gentle reminder to Reviewer H7vm**
>
> Dear Reviewer **H7vm**,
>
> We thank you again for your insightful comments and your time reviewing our paper.
>
> We would appreciate it if you could confirm that our responses address your concerns. We are happy to answer more if you have any remaining concerns or questions.
>
> Best regards,
>
> Authors

---

> ### Author Response · Authors · 2022-12-08
> **A kindly reminder to Reviewer H7vm**
>
> Dear Reviewer **H7vm**,
>
> Thank you again for your precious time!
>
> We kindly ask the reviewer to reassess the paper in light of this comment if it clears things up. We are happy to answer more if you have any remaining concerns or questions.

---

### Decision · Program_Chairs · 2023-01-20

**Decision:**

Accept: poster

**Justification For Why Not Higher Score:**

This paper provides a nice analysis and method for budgeted training, which would be of interest to the community. The paper could be accepted as spotlight although I don't believe that the scores and strength of the paper necessarily justifies that.

**Justification For Why Not Lower Score:**

Overall, all of the reviewers appreciated this paper's contributions, especially after the rebuttals which included significant additional experiments showing the generality of the method. The two positive reviewers confirmed that they vote for acceptance.

**Metareview: Summary, Strengths And Weaknesses:**

This paper looks at the interesting budgeted training problem: How to train models under a given budget (training time/compute) to achieve a good accuracy under such constraints. The paper takes the perspective of identifying redundancies in the attention heads, MLPs, and visual tokens of Vision Transformer architectures and uses this insight to suggest a progressive growing mechanism used across a three-phase training regime.

The reviewers all appreciated the writing clarity, problem setting (relevant given the current growth of compute in training these models), analysis demonstrating redundancy in different components, and evaluation across multiple architectures/ablations showing benefits across different budgets. Some suggestions were made to improve various aspects of the paper, including improving the narrative of the motivation to include discussion of other potential approaches and the angle this paper takes on the problem, comparison to some of the other approaches (e.g. pruning), comparison of different learning schedules, and demonstration of generalization across additional downstream tasks. The authors provided an extensive rebuttal, including significant new experiments addressing these, showing that the method is indeed general to other vision tasks.

  Overall, the problem of budgeted training is extremely important from a variety of practical, research, and societal aspects and this paper takes an interesting perspective that is well-motivated through analysis to yield an effective method. Therefore I recommend acceptance. I recommend that the authors update the manuscript to include all of the discussion items, especially the new experiments, in the camera-ready version, as they significantly add to the demonstrated effectiveness and generality of the method.

**Note From Pc:**

if the above contains the word "oral" or "spotlight" please see: "oral" presentation means -> notable-top-5% and "spotlight" means -> notable-top-25%. As stated in our emails, we are disassociating presentation type from AC recommendations